# Multi-Temporal Variabilities of Evapotranspiration Rates and Their Associations with Climate Change and Vegetation Greening in the Gan River Basin, China

**Meng Bai [1,3]** , **Bing Shen [2,\*], Xiaoyu Song [2], Shuhong Mo [2], Lingmei Huang [2] and Quan Quan [2]**

1. Institute of Geographic Sciences and Natural Resources Research, Chinese Academy of Sciences, Beijing 100101, China; benjamin0054@163.com
2. State Key Laboratory of Eco-hydraulics in Northwest Arid Region, Xi'an University of Technology, Xi'an 710048, China; songxy@xaut.edu.cn (X.S.); moshuhong@xaut.edu.cn (S.M.); huanglm@xaut.edu.cn (L.H.); qq@xaut.edu.cn (Q.Q.)
3. University of Chinese Academy of Sciences, Beijing 100049, China
* Correspondence: shenbing@xaut.edu.cn

**Abstract:** Understanding the spatial-temporal dynamics of evapotranspiration in relation to climate change and human activities is crucial for the sustainability of water resources and ecosystem security, especially in regions strongly influenced by human impact. In this study, a process-based evapotranspiration (ET) model in conjunction with the Global Land Surface Satellite (GLASS) LAI dataset was used to characterize the spatial-temporal pattern of evapotranspiration from 1982 to 2016 over the Gan River basin (GRB), the largest sub-basin of the Poyang Lake catchment, China. The results showed that the actual annual ET (ETa) weakly increased with an annual trend of 0.88 mm year$^{-2}$ from 1982 to 2016 over the GRB, along with a slight decline in annual potential ET (ETp). On an ecosystem scale; however, only the evergreen broadleaved forest and cropland presented a positive ETa trend, while the rest of the ecosystems demonstrated negative trends of ETa. Both correlation analysis and sensitivity analysis revealed a close relationship between ETa inter-annual variability and energy availability. Attribution analysis illustrated that contributions of climate change and vegetation greening on the ETa trend were −0.48 mm year$^{-2}$ and 1.36 mm year$^{-2}$, respectively. Climate change had a negative impact on the ETa trend over the GRB. However, the negative effects have been offset by the positive effects of vegetation greening, which mainly resulted from the large-scale revegetation in forestland and agricultural practices in cropland. It is concluded that large-scale afforestation and agricultural management were the main drivers of the long-term evolution of water consumption over the GRB. This study can improve our understanding of the interactive effects of climate change and human activities on the long-term evolution of water cycles.

**Keywords:** evapotranspiration; LAI; process-based ET model; sensitivity analysis; Poyang Lake basin

---

## 1. Introduction

Climate change and human activities have altered the global hydrological cycle at multiple spatiotemporal scales. Due to the tremendous heterogeneity in climate and geographical conditions in different regions, responses of the hydrological cycle to the environmental changes are region-dependent. With climate warming, the annual runoff of major rivers in north China has shown a decreasing trend, aggravating the existing water shortage in this region [1–3]. However, for most rivers in south China, the impacts of seasonal pattern changes in hydrological variables are more prominent than those of the

inter-annual changes [4]. For example, due to the seasonal asymmetry of the changes in precipitation, the upper Yangtze River basin has witnessed both an increase in summer runoff and a significant decrease in autumn runoff over the past decades, which posed a certain threat to the water security and ecological environment [5]. Further, climate variability has been revealed to increase security risks with respect to water supply and food production (Intergovernmental Panel on Climate Change (IPCC) Fifth Assessment Report). Hence, understanding the eco-hydrological processes and the associated impacts on water budget components is essential for water security, hydrological prediction, crop irrigation schedule, and ecosystem conservation, which are indispensable for practices of integrated catchment management and water resources assignment [6].

Poyang Lake is the largest freshwater lake in China and located in the middle reaches of the Yangtze River basin. The catchment of Poyang Lake plays an important ecological and hydrological role in the middle and lower reaches of the Yangtze River and also acts as one of the major agricultural production areas in China. However, influenced jointly by the climatic and non-climatic factors, the hydrological regimes in the Poyang Lake basin (PLB) are changing significantly. For example, the rise in frequency and severity of floods has been observed in the PLB in recent decades [7,8], which is partly attributed to the increased fluctuation of the warm-season rainfall [9]. Meanwhile, more frequent droughts in the PLB have been observed since 2000, especially during autumn and summer [10,11], with an enhanced influence in terms of duration, frequency, intensity, and severity [12]. These prominent changes in the hydrological situation have raised the concern about what natural and anthropogenic factors are and how these factors have contributed to the changes. Sun et al. [13] showed that the streamflow at four hydrological stations of the PLB exhibited an increasing trend between 1961–2000, and that the increase in precipitation and decrease in potential evapotranspiration were the main contributors to the streamflow increment. Based on a coupled water and energy budget analysis, Ye et al. [14] confirmed the dominant role of climate change in variations of mean annual streamflow in the PLB and found that the relative effects of climate change and human activities varied among sub-catchments, as well as the whole catchment during different decades. Zhang et al. [15] further discovered that although climate change dominated the annual change of streamflow, human activities also played a key role in the streamflow variation during some months. As the main type of land-cover change in recent decades, afforestation and revegetation (i.e., the Mountain-River-Lake watershed management program (MRL) and the Grain for Green (GFG) project) are considered to be the dominant ways that humans influence the hydrological processes in PLB [10,16]. The MRW program was launched in the early 1980s and the GFG project was implemented in the late 1990s. Engineering measures of these projects mainly include forestation, natural forest conservation, and returning mountainous farmland to forest, etc. [16]. These programs have improved vegetation growth and contributed to vegetation greening. Huang et al. [16] reported that the reforestation in the PLB had promoted the forest coverage from 33.1% in 1983 to 60.5% in 2011. Meanwhile, improvements in agricultural management practices (renewals of cultivars, irrigation facility improvement, fertilizer application) also resulted in an obvious increase in the coverage and greenness of the cropland in the PLB [10]. These human-induced changes of the underlying surface are expected to have exerted great impacts on the hydrological processes in the PLB [11,14,17]. On the basis of model experiments, Guo et al. [7] concluded that the increase of forest cover reduces streamflow in wet seasons and increases it in dry seasons; thus, reducing flood potentials in the wet season and drought severity in the dry season. However, Tang et al. [10] reported that during continuing and intensifying droughts, increased vegetation greenness could cause or aggravate water conflicts in sub-watersheds with high forest cover and high human water demands. Overall, the long-term evolution of eco-hydrology in response to climate change and vegetation greening in the PLB is still not well understood, and the relative role of climate variability and human activities in driving the multiple spatiotemporal eco-hydrological processes is yet to be investigated.

Previous studies on hydrological changes in the PLB mainly concentrated on the changes in streamflow and their associations with climatic and anthropogenic factors. However, these runoff-focus studies are usually difficult to uncover the differences in hydrological responses among

terrestrial ecosystems, since they were generally conducted on a catchment scale. Essentially, terrestrial evapotranspiration (ET) is a better indicator of hydrological change than streamflow, since the runoff process is intermittent and influenced by various factors, whereas ET occurs every day and its evolution is relatively stable. Also, ET plays a vital role in connecting water, energy, and carbon cycles in the terrestrial ecosystem [18], and its long-term tendency of ET may be regarded as a critical indicator of regional water cycle intensification. Terrestrial ET is considerably affected by climate variabilities through a set of coupled physical and physiological processes. Climate variabilities also exert significant impacts on the ecological and hydrological processes, which interact crossing scales in complex ways and are closely linked to the types of ecosystems. However, ecosystem resilience may maintain the plant water use efficiency through modifying vegetation dynamics (phenology, photosynthesis, canopy density, etc.), which to some extent, has stabilized the function of the ecosystem exposed to climate variations [19,20]. Eco-hydrological responses to environmental change vary with vegetation type or ecosystem, due to differences in the physiological structures among vegetation types [21–24]. On the other hand, vegetation dynamics may exert significant impacts on the hydrological processes, such as transpiration, streamflow, and groundwater recharge [25,26]. Hence, a careful and in-depth investigation of the ET spatiotemporal changes over different ecosystem types and their driving factors can improve our abilities to predict land surface-atmosphere interactions and terrestrial ecosystems dynamics in response to climate change and land-cover changes.

Accurate estimation of water vapor flux is important to evaluate the eco-hydrological responses to climate variability and land use/cover changes on a basin or regional basis. However, in recent decades, direct ET measurements have only been available at tower sites with the eddy covariance technique or the Bowen ratio system [27]. ET information at large scale is still derived indirectly, along with considerable uncertainties. In many cases, the basin-scale annual ET is only estimated as a residual of precipitation minus the stream discharge, whose accuracy is questioned due to the lack of information about the changes in the terrestrial water storage (TWS). However, this shortcoming has been overcome by the Gravity Recovery and Climate Experiment (GRACE) data, which provided the spatiotemporal variations of TWS anomalies over the global surface since 2002 [28]. Due to tremendous heterogeneity in the vegetation conditions, soil texture, geomorphology, hydrologic, and climatic forces, obtaining the detailed information of the spatial distribution of ET at a large scale seemed to be out of reality a few decades ago. However, the advent and development of remote sensing (RS) technology made it possible to directly retrieve the large-scale land surface characteristics from the remotely sensed information [29–31]. For example, visible and near-infrared spectral reflectance and their combinations are widely used to detect the spatiotemporal variations of vegetation cover fraction and leaf area index [32–34], which are the main factors regulating the spatial pattern of ET. On this basis, process-based land surface models or eco-hydrological models that integrate vegetation information remotely sensed by frequently revisiting satellites and ground climate data have been developed to predict the spatiotemporal variations of ET at multiple spatiotemporal scales [35–38]. In these models, the prevailing and physically-based ET schemes, such as the Penman–Monteith (P–M) model and the Priestley–Taylor (P–T), are commonly employed. Although some ET datasets generated by the models (e.g., MODIS-ET) have been validated at a global scale, their performances at regional scales are yet to be improved [29,37,39]. For the prediction of regional or basin ET, more elaborate schemes and more careful calibration should be used to account for the ET variations at a smaller spatiotemporal scale.

The purpose of this study is to investigate the evolution of eco-hydrological processes and the driving mechanism in the PLB during the past several decades. By using a process-based evapotranspiration model integrated with the Global land surface satellite (GLASS) LAI products at 5 km resolution from 1982 to 2016, the spatiotemporal pattern of ET and the driving forces in the Gan River basin (GRB), the largest sub-basin of the PLB, are explored. The following issues are revealed: (1) multi-temporal variabilities of ET over the GRB and over ecosystems during the past several decades; (2) driving forces that dominated ET multi-temporal variabilities; (3) contributions of climate change

and human activities (mainly referring to the land use/cover change, agricultural management, etc.) on the interannual trend of ET over ecosystems and over the GRB.

## 2. Method and Materials

### 2.1. Study Basin

Located on the south bank of the middle reaches of the Yangtze River, the Gan River basin (GRB) is the largest sub-basin of the Poyang lake and also one of the principal southern tributaries of the Yangtze River, with an approximate area of 83,500 km$^2$ (Figure 1a). Landforms in the basin are relatively complex, with the central and south parts being dominated by mountainous regions and subordinate hilly areas, and the lower basin featured with alluvial plains [40]. Hence, the altitude span of the GRB is fairly large, ranging from 2045 m in the southwest to 12 m in the north. The total length of the Gan River is 815 km, and the multiannual average discharge was $6.79 \times 10^{10}$ m$^3$ over the period 1953–2014 at Waizhou station—the outlet of the GRB, accounting for 50% of the total runoff within Poyang Lake Basin. Land use derived from the Landsat ETM+ in 2015 showed that forest land is dominated in the GRB, accounting for 65.2% of the total area (Figure 1b). Agricultural land, pasture, urban areas, and open water cover 24.9%, 5.5%, 2.3%, and 1.6% of the basin, respectively. The double rice cropping system (early rice and later rice) prevails over the plain areas of the GRB.

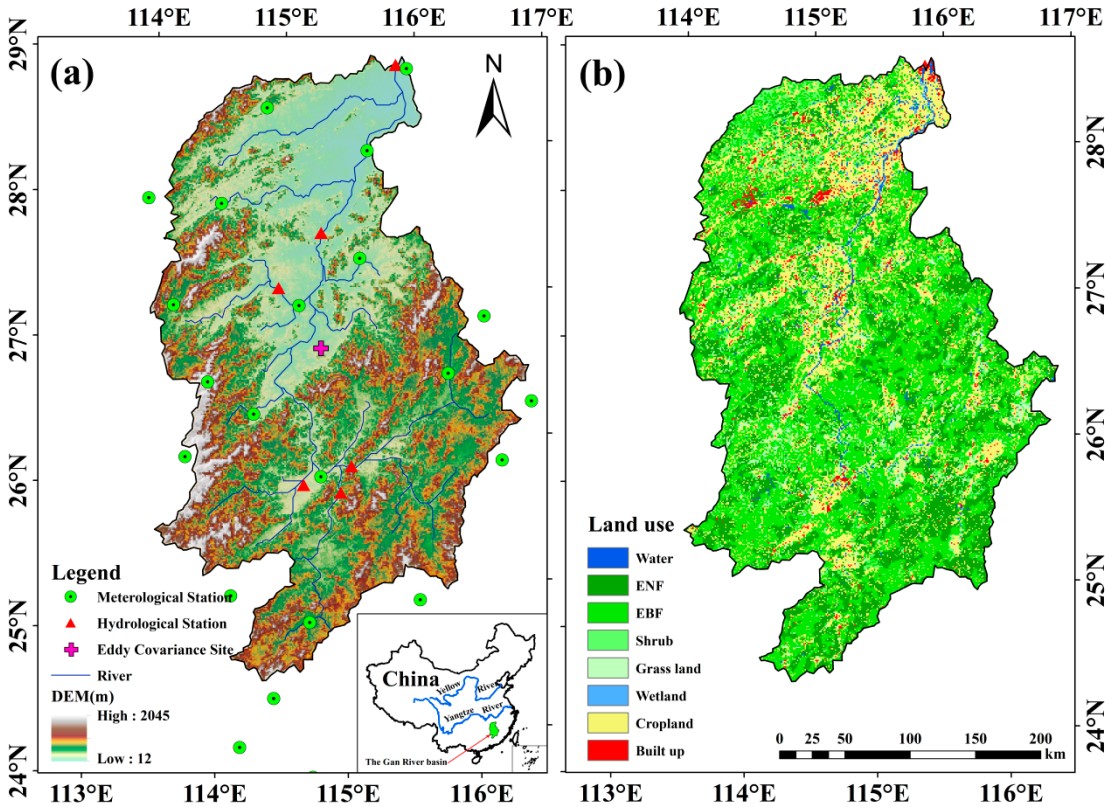

**Figure 1.** (**a**) Location of the Gan River basin and the distribution of hydrological stations, meteorological stations and the flux tower; (**b**) land use/cover map of the Gan River basin in 2015 (ENF: evergreen needle-leaved forest; EBF: evergreen broad-leaved forest).

The GRB has a humid subtropical climate mainly governed by the East Asian monsoon, with an average annual temperature ranging from 16.6 °C to 20.3 °C, and an average annual precipitation from 1450 mm to 1770 mm. The precipitation shows strong seasonality, with 53.2% of the annual amounts concentrated in the wet season (March to June) but only 28.2% in the dry season (July to October) (Figure 2). Therefore, the wet season is more prone to flooding, while the dry season is vulnerable to

drought [41]. Meteorological observations showed a warm-wet trend in climate over the past decades, with an annual trend of 0.014 °C year$^{-1}$ for the air temperature and 4.136 mm year$^{-1}$ for precipitation (Table 1). Meanwhile, both wind speed and relative humidity exhibited a negative trend, in line with that of other parts of eastern China [42]. The seasonal variations of streamflow are principally driven by precipitation, showing an increase in the first half of the year, and then a decrease in the second half of the year with a sharp decline from June to August.

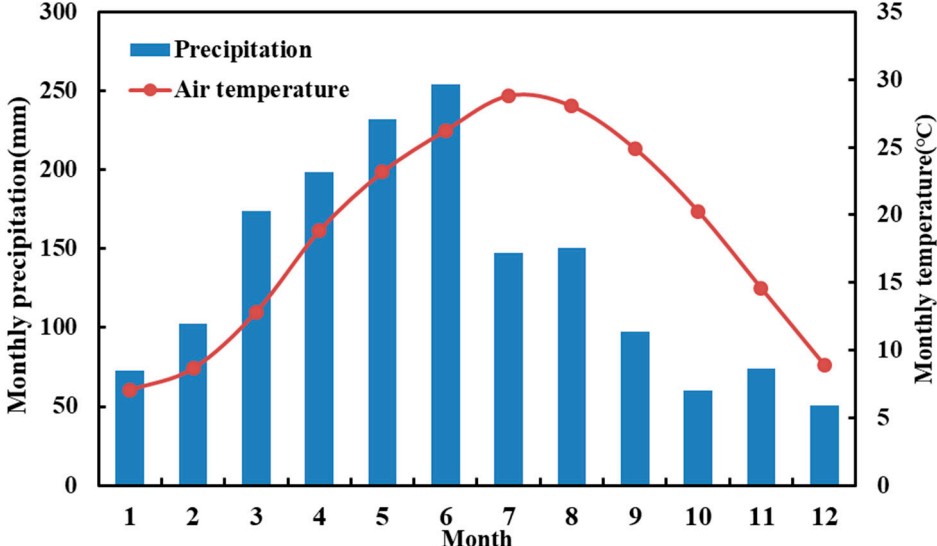

**Figure 2.** Monthly precipitation and air temperature in the Gan River basin (GRB) averaged over the period of 1982–2016.

**Table 1.** Inter-annual trends of the annual and monthly air temperature (T), precipitation (P), wind speed (U), and sunshine duration (SD) from 1982 to 2016 over the Gan River basin (GRB).

| Climatic Variables | Jan | Feb | Mar | Apr | May | Jun | Jul | Aug | Sep | Oct | Nov | Dec | Annual |
|---|---|---|---|---|---|---|---|---|---|---|---|---|---|
| T (°C year$^{-1}$) | −0.003 | −0.027 | 0.067 ** | 0.041 * | 0.008 | 0.012 | 0.001 | −0.004 | 0.017 | 0.028 | 0.015 | 0.01 | 0.014 |
| P (mm year$^{-1}$) | 0.17 | −2.4 ** | −1.332 | −0.522 | 1.975 | 1.173 | 1.231 | 1.36 | −0.271 | −0.858 | 2.209 * | 1.401 | 4.136 |
| U (m s$^{-1}$ year$^{-1}$) | −0.013 ** | −0.011 ** | −0.014 ** | −0.009 ** | −0.011 ** | −0.013 ** | −0.014 ** | −0.004 | −0.009 ** | −0.01 ** | −0.009 ** | −0.008 ** | −0.01 ** |
| SD (h year$^{-1}$) | −0.235 | 1.004 | 1.002 * | 0.281 | −0.743 | −0.869 | −0.764 | −1.048 | −0.355 | 0.069 | −0.933 | −0.891 | −3.48 |

Significance level: * ($p < 0.05$) ** ($p < 0.01$).

## 2.2. Model Description

The model used in this study is a daily-scale process-based ET model. It adopts a dual-source scheme on the basis of the Penman–Monteith equation. In this model, net radiation at the canopy top ($R_n$, MJ/(m$^2$·d)) is further partitioned into energy for the canopy and that for the soil ($R_{nc}$ for the canopy and $R_{ns}$ for the soil). Correspondingly, total ET is calculated as the sum of canopy transpiration ($E_c$), soil evaporation ($E_s$), and the canopy interception evaporation ($E_i$) using the Penman–Monteith type equation [43]. $E_c$ and $E_s$ (mm/d) are estimated as follows:

$$E_c = \frac{1}{\lambda} \frac{\Delta R_{nc} + F_r \rho C_p D / r_{ac}}{\Delta + \gamma \left(1 + \frac{r_c}{r_{ac}}\right)} \tag{1}$$

$$E_s = \frac{1}{\lambda} \frac{\Delta (R_{ns} - G) + (1 - F_r) \rho C_p D / r_{as}}{\Delta + \gamma \left(1 + \frac{r_s}{r_{as}}\right)} \tag{2}$$

where $\lambda$ is the latent heat of vaporization of water (MJ/kg); $\Delta$ is the slope of the saturated vapor pressure curve versus air temperature (hPa/K); $G$ is the soil heat flux (MJ/(m$^2$·d)); $F_r$ is the fractional vegetation cover and retrieved with the remote sensing vegetation index [44]; $r_c$ and $r_s$ are the bulk canopy stomatal

resistance and soil surface resistance, respectively (s/m); $r_{ac}$ and $r_{as}$ are the aerodynamic resistances on the canopy and soil surface, respectively (s/m); $\gamma$, $\rho$, $C_p$, and $D$ represent the psychrometric constant (hPa/K), air density (kg/m$^3$), specific heat capacity of air (MJ/(kg·K)), and vapor-pressure deficit (hPa), respectively.

Net radiation at the top of the canopy ($R_n$) is estimated using the empirical relationships recommended by the Food and Agriculture Organization (FAO) [43], in which $R_n$ is calculated as the difference between net shortwave radiation ($R_{ns}$) and net long-wave radiation ($R_{nl}$), namely:

$$R_n = R_S - R_L \tag{3}$$

$$R_S = (1 - \alpha)\left(0.25 + 0.5\frac{n}{N}\right)R_O \tag{4}$$

$$R_L = \left(0.1 + 0.9\frac{n}{N}\right)\left(0.34 - 0.14\sqrt{e_a}\right)\sigma(T_a + 273)^4 \tag{5}$$

where $R_S$ and $R_L$ are the incoming net shortwave radiation and the outgoing net long-wave radiation (MJ/d), respectively; $R_O$ is the incoming solar radiation at the top of the atmosphere (MJ/d); $\alpha$ is the land surface albedo; $n$ and $N$ are the actual and potential sunshine durations, respectively; $e_a$ is the air vapour pressure (hPa); $\sigma$ is the Stefan–Boltzmann constant; and $T_a$ is the daily average air temperature. The land surface albedo ($\alpha$) used for calculating $R_{ns}$ is related to leaf area index [45] as:

$$\alpha = \alpha_m - (\alpha_m - \alpha_s)\exp(-0.56\,LAI) \tag{6}$$

where $\alpha_m$ and $\alpha_s$ are the albedo corresponding to the "closed" canopy and the bare soil, respectively. According to Liu et al. [46], surface albedo in barren land varied from 0.194 to 0.250 during an average year. Therefore, $\alpha_s = 0.25$ is used in this study given that $\alpha_s$ denotes the surface albedo in an ideal bare surface. $\alpha_m$ is from the literature and varies with vegetation types [36]; $LAI$ is the satellite-based leaf area index. $R_n$ is further partitioned into $R_{nc}$ and $R_{ns}$ using a layer approach of Beer's law, namely, $R_{nc} = R_n e^{-k_c LAI}$ and $R_{ns} = R_n - R_{nc}$, in which $k_c$ is the extinction coefficient for net radiation.

Bulk canopy stomatal resistance ($r_c$) is estimated using the approach proposed by Jarvis [47], in which $r_c$ is assumed to be a function of vegetation type (denoted by the minimum leaf stomatal resistance under the optimal condition) and the environmental conditions. The soil surface resistance ($r_s$) is related to the soil water content near the soil surface layer. The aerodynamic resistance on the canopy surface ($r_{ac}$) is calculated using the scheme of Choudhary et al. [48], which is a function of canopy structure (represented by the characteristic length of leaf width), leaf area index, and wind speed at the canopy height. The aerodynamic resistance on the soil surface ($r_{as}$) is estimated using the approach proposed by Guan et al. [49], which is a transformation of the scheme of Campbell et al. [50].

Soil water movement is described using a discrete form of Richards' equation. The root zone (soil depth within 1.6 m) is divided into three layers in the vertical direction. Soil water exchange between layers is governed by Darcy's law in which the soil hydraulic parameters are estimated using the scheme of Clapp et al. [51], and the changes of soil water in each layer are simulated based on a water-balance equation. Rainfall infiltration is an external forcing of soil water movement and is estimated using the water storage capacity curve method on a daily scale.

In this study, we assume that the long-term trend of LAI (vegetation greening) may represent the effects of human activities on crop and natural ecosystems. Therefore, human activities are mainly incorporated into the model via LAI, which is used to retrieve the vegetation dynamics and land surface characteristics. Irrigation is assumed as the main way of anthropogenic water utilization over the GRB. During the rice-growing period (late April to early November), the volume of irrigation water at different growth stages was set according to the Agricultural Irrigation Water Quota of the Jiangxi Province.

*2.3. Data*

2.3.1. Model Input Data

The model input data include Digital elevation model (DEM), land use/cover, vegetation characteristic, soil physical properties, and atmospheric forcing variables. The DEM data originated from the SRTM 90 m Digital Elevation Database v4.1 (http://srtm.csi.cgiar.org/srtmdata/) and was used to derive the topography map of the GRB. Also, it was used for spatial interpolation of climatic variables.

A land-use/cover dataset (LU, spanning from 1980 to 2015 and being available in every five years) and a vegetation type dataset (VT) were used to generate the land use map of the GRB. The LU classification was derived from the Landsat TM/ETM+ images for each period, and the VT dataset was originated from the 1:1,000,000 vegetation map of China [52]. These datasets were all provided by the Data Center for Resources and Environmental Sciences, the Chinese Academy of Sciences (RESDC) (http://www.resdc.cn), and were available with a spatial resolution of 1 km. The land use map of the GRB was derived from a combination of the LU and VT, in which the overall pattern of land use in the LU was preserved while the more detailed vegetation type information in the VT was also incorporated.

The Global Land Surface Satellite (GLASS) LAI product, generated from the Moderate-Resolution Imaging Spectroradiometer (MODIS) and Carbon Cycle and Change in Land Observational Products from an Ensemble of Satellites (CYCLOPES) LAI products as well as MODIS reflectance products [53], was used to retrieve canopy phenology and land surface characteristics. This dataset spans from 1982 to 2016 with an eight-day temporal composite at 5 km spatial resolution. For quality control, the LAI data were first corrected with Savitzky–Golay (S–G) filter, which has been confirmed to be a reliable way to remove the contamination by cloud and abrupt points [54]. Then, the eight-day data were interpolated to daily values using the Lagrange polynomial method.

Soil texture data were digitized from a 1:1,000,000 scale map [55] in which the surface soil texture was classified into 11 types, according to the fractions of sand, silt, and clay. The parameters of soil porosity and saturated hydraulic conductivity are estimated as in Bonan [56].

Daily meteorological data (precipitation, air temperature, air pressure, relative humidity, wind speed, and sunshine duration) at 35 stations inside and around the GRB were employed to generate the spatial regime of atmospheric forcing with the spatial resolution of 5 km by gradient inverse distance square (GIDS) method [57], in which the effects of terrain, latitude, and longitude were considered with multivariate regressive analysis.

All of the geographic information and remote sensing data were projected into the Lambert Azimuthally Equal-Area Projection with a spatial resolution of 5 km. A summary of the datasets used in this study can be seen in Table 2.

**Table 2.** Summary of the datasets used in this study.

| Datasets | Spatial Resolution | Temporal Resolution | Time Span | Source |
|---|---|---|---|---|
| Digital Elevation Model (DEM) | 90 m × 90 m | | | http://www2.jpl.nasa.gov/srtm/ |
| Land use/cover data | 1 km × 1 km | 5\five-year | 1980–2015 | http://www.resdc.cn |
| Vegetation type map | 1 km × 1 km | | | http://www.resdc.cn |
| Global land surface satellite (GLASS) LAI | 5 km × 5 km | eight-day | 1982–2016 | http://glass-product.bnu.edu.cn/ |
| Soil texture data | 1:1,000,000 scale | | | http://geodata.pku.edu.cn |
| Daily meteorological observations | N/A | daily | 1980–2016 | http://data.cma.cn/ |
| Gravity Recovery and Climate Experiment (GRACE) RL 05 data | 1.0° × 1.0° | Monthly | 2002–2015 | http://www2.csr.utexas.edu/grace/ |
| Eddy covariance flux measurements at Qianyanzhou station | | daily | 2003–2005 | http://www.chinaflux.org/ |

2.3.2. Data for Model Validation

Eddy covariance flux measurements spanning from 2003 to 2005 at Qianyanzhou (QYZ) station (115°03′29.2″ E, 26°44′29.1″ N) were used for the validation of daily-scale ET simulations. The land cover at QYZ station is the evergreen needle-leaved forest, which is one of the main vegetation types in GRB. The regional ET at monthly and annual scales was validated with the water-balance derived ET at the basin scale, which is expressed as the residual of the water-balance equation, namely:

$$ET = P - R - \Delta S \tag{7}$$

where $P$ and $R$ represent the monthly or annual precipitation and runoff, respectively; $\Delta S$ is the change in the terrestrial water storage (TWSC) during the corresponding period. The monthly runoff data at the Waizhou hydrological station were collected from the Hydrologic Yearbook of China. The soil water content simulated by this model was not used as the representation of water storage, because the terrestrial water storage (TWS) consists of not only the soil water but also the stored surface water and the groundwater storage. Instead, the Gravity Recovery and Climate Experiment (GRACE) data, which represent the terrestrial water storage anomalies (TWSA) over the global surface, were used in this study (https://grace.jpl.nasa.gov/data/get-data/monthly-mass-grids-land/).

The GRACE satellites acquired monthly terrestrial water storage anomalies by monitoring the spatial-temporal variations of the Earth's surface mass, which have been widely used to derive the spatial-temporal patterns of TWS [58], groundwater [59], ET [60,61], etc. In this study, the GRACE RL05 data released by Center for Space Research of the University of Texas (CSR) were used to derive the monthly/annual ET time series, and thus, provide validation for the modeled ET on the basin scale (http://www2.csr.utexas.edu/grace/). This dataset covers the period from April 2002 to December 2015 with a spatial resolution of $1.0° \times 1.0°$. In the generation of the GRACE RL05 product, a de-striping filter and a 300-km Gaussian filter were applied to the primitive GRACE data to minimize errors associated with the correlated noise and instrumental noise, which are manifested as north-south striping patterns in the spatial domain. In this study, the data were re-corrected by the scaling factor approach to restore signal losses arising from the sampling and post-processing of GRACE data (i.e., leakage errors) [62]. The scaling factor approach uses the TWS output from land surface models (LSMs) to generate scaling factors for correcting the errors in GRACE data [63]. In this study, the scaling factors were generated based on an LSM in the Global Land Data Assimilation System (GLDAS), i.e., the Variable Infiltration Capacity (VIC) [63]. Although the GRACE data provide an unprecedented opportunity to validate or constrain the hydrological model in combination with streamflow observations [64,65], the coarse spatial resolution may introduce uncertainties in the estimation of TWSA for middle-scale basins such as the GRB, because considerable proportions of GRACE grids within the GRB are across the basin boundary (Figure 3a). To this end, we generated a finer resolution ($0.25° \times 0.25°$) time series of TWSA (Figure 3b) over the GRB using a model-based downscaling approach proposed by Wan et al. [66], in which the finer spatial patterns of the model-based TWSA are incorporated into the GRACE TWSA data, while the original spatial variations of GRACE TWSA are also preserved. In this study, a $0.25° \times 0.25°$ hydrological variable dataset generated by the Variable Infiltration Capacity (VIC) model was employed to downscale the GRACE data [67]. Figure 3 shows the inter-annual trends of the original- and downscaled- GRACE TWSA in June from 2000 to 2012. As shown in Figure 3, the downscaled GRACE data exhibited a similar spatial pattern but provided more detailed spatial information than the original GRACE data.

The downscaled gridded GRACE data were then aggregated to the basin-scale TWSA. To better interpret the inter-annual variations of ET in the GRB, a long-term basin-scale TWSA time series from 1982 to 2015 was reconstructed by setting up a multiple regression model, with annual precipitation and annual runoff as the predictor variables and TWSA as the response variable. The regression model came out to be strongly significant ($R^2 = 0.86$, $p < 0.01$). As shown in Figure 4, the reconstructed TWSA

and GRACE TWSA showed similar inter-annual variations from 2003–2015, both exhibiting favorable agreement with precipitation anomalies at the inter-annual timescales.

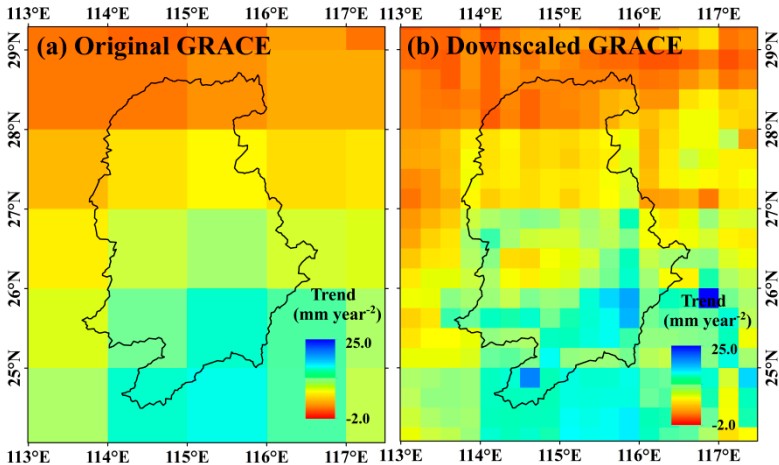

**Figure 3.** Comparisons of inter-annual trends of original- and downscaled- Gravity Recovery and Climate Experiment (GRACE) data in June during 2002–2012. (**a**) Original GRACE; (**b**) Downscaled GRACE.

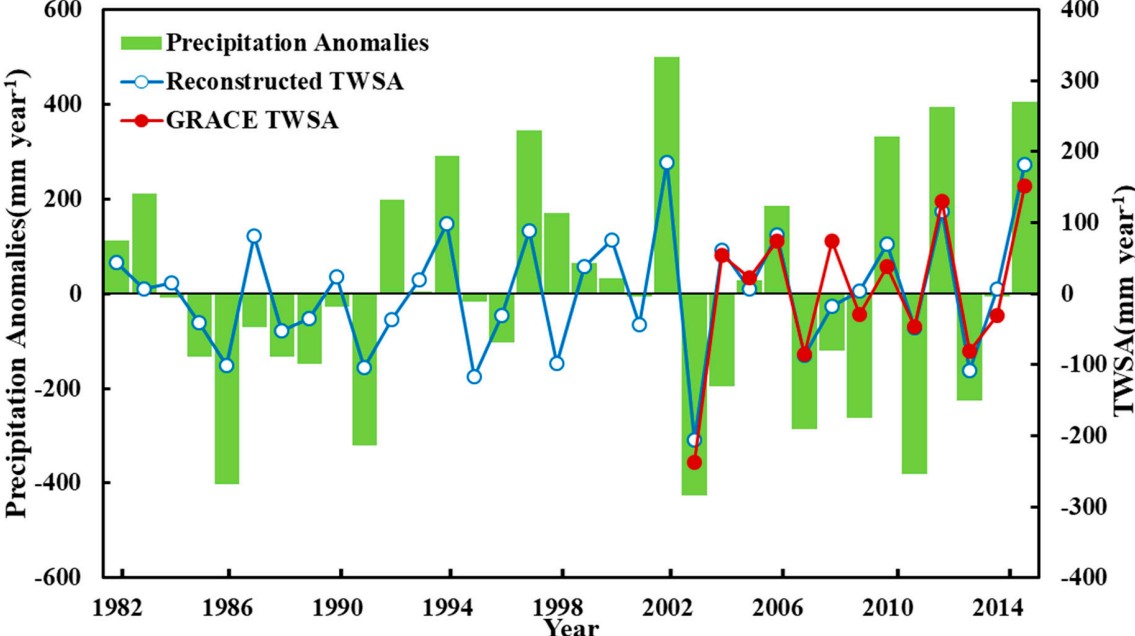

**Figure 4.** Annual variations of GRACE-derived and reconstructed Terrestrial Water Storage Anomalies (TWSA). The green bars show annual precipitation anomalies relative to the average annual precipitation during 1982 to 2015.

*2.4. Methods*

2.4.1. Analysis Methods

The ET of the GRB was simulated from 1980 to 2016 with a 5 km spatial resolution and daily time step, wherein, the first two years were taken as the warming-up period. To evaluate the relationship between two variables, Pearson's correlation analysis was performed with the coefficient of determination ($R^2$) to describe its level of significance. To trace the inter-annual variation of ET, LAI, and climate variables, the long-term annual trends of these variables were determined by the slope of the simple linear regression model. In addition, the partial correlation analysis was used to

investigate relationships between ET, LAI, and climate variables, considering that all the variables are associated with each other. The partial correlation analysis was conducted with the linear trends of all the variables removed in order to focus on the relationship of the year-to-year variations. The statistical significance of a linear trend or a linear correlation was assessed using the t-test with a confidence interval of 95% (i.e., $p < 0.05$).

### 2.4.2. Sensitivity Analysis Method

Sensitivity/Elasticity theory has been widely adopted in analyses of hydrological variables in recent years [68–70]. The essence of this theory is to reveal the response rates of the dependent variable to multiple independent factors by setting up a linear regression model with variations. The sensitivity/elasticity coefficient represents the response rate of the dependent variable, responding to the variation of each independent variable. In this study, the long-term response of ET to climate variability and vegetation greening was revealed by the sensitivity method.

For the soil-plant-atmosphere continuum system, the water vapor flux (*ET*) between the land surface and the atmosphere is determined by multiple climatic drivers (Precipitation (*P*), Temperature (*Ta*) and sunshine duration (*SD*), etc.) and the vegetation condition (represented by *LAI* in this study), namely,

$$ET = f(P, Ta, SD, LAI, \ldots) \tag{8}$$

The changes in ET in response to the small changes of multiple drivers can be expressed approximately with the first-order Taylor expansion as:

$$\Delta ET \cong \frac{\partial f}{\partial P} \Delta P + \frac{\partial f}{\partial Ta} \Delta Ta + \frac{\partial f}{\partial SD} \Delta SD + \frac{\partial f}{\partial LAI} \Delta LAI + \ldots \tag{9}$$

where $\Delta ET$, $\Delta P$, $\Delta Ta$, $\Delta SD$, and $\Delta LAI$ represent absolute changes in ET, precipitation, air temperature, sunshine duration, and LAI, respectively; $\frac{\partial f}{\partial P}$, $\frac{\partial f}{\partial Ta}$, $\frac{\partial f}{\partial SD}$, and $\frac{\partial f}{\partial LAI}$ denote the response rates of ET responding to changes in precipitation, air temperature, sunshine duration, and LAI, respectively.

The values of these response rates depend on the unit of the independent variables. In other words, they are dimension-dependent. To eliminate the influence of dimensions, the absolute changes in the variables were replaced by the corresponding relative changes (except for Ta, for which the sensitivity to a perturbation of 1K is more useful), and Equation (3) was rewritten as:

$$\frac{\Delta ET}{\overline{ET}} = S_P \frac{\Delta P}{\overline{P}} + S_{Ta} \Delta Ta + S_{SD} \frac{\Delta SD}{\overline{SD}} + S_{LAI} \frac{\Delta LAI}{\overline{LAI}} + \ldots \tag{10}$$

where $\overline{ET}$, $\overline{P}$, $\overline{SD}$, and $\overline{LAI}$ refer to the long-term averaged values of ET, precipitation, air temperature, sunshine duration, and LAI, respectively. $S_P$, $S_{SD}$, $S_{LAI}$ and $S_{Ta}$ are the sensitivity coefficients of ET in response to changes in precipitation, sunshine duration, LAI and air temperature, respectively. Then, contribution of one single variable to ET change could be expressed as (take LAI as an example):

$$Con(LAI) = S_{LAI} \frac{\Delta LAI}{\overline{LAI}} \overline{ET} \tag{11}$$

where $Con(LAI)$ is the contribution of LAI change ($\Delta LAI$) to ET change. If $\Delta LAI$ indicates the inter-annual trend of LAI, $Con(LAI)$ should be interpreted to the contribution of LAI trend to ET trend.

The conceptual scheme and methodology flow chart for this study can be seen in Figure 5.

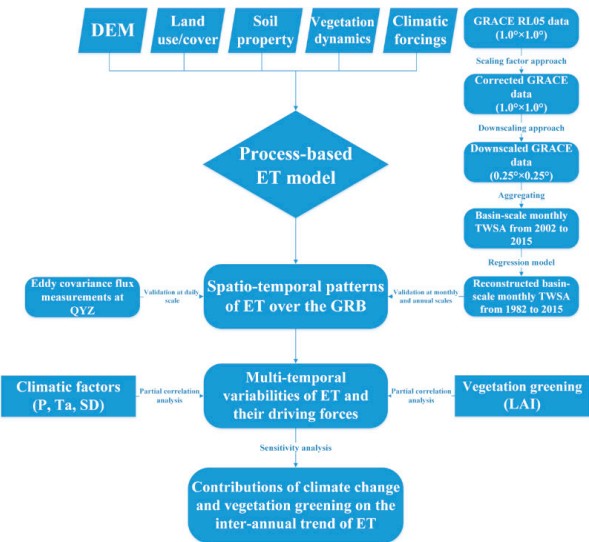

**Figure 5.** Flow chart of the methodology employed in this study.

## 3. Results

### 3.1. Model Validation

The simulated daily ET rates in the pixel, where QYZ site is located, were validated with measurements of eddy covariance at QYZ site, a needle-leaved forest field within the GRB. In general, agreements between measurements and predictions are quite satisfactory. As shown in Figure 6, the simulated and measured daily ET exhibited a significant correlation, with the coefficient of determination ($R^2$) of 0.63 ($p < 0.01$), and the root mean square errors (RMSE) of 0.80 mm/d. This indicates that the simulated daily ET basically traced the seasonal fluctuations of the field measurements. It is noted that the model generated more severe errors in overprediction than underprediction, especially in the 2–3 mm/d range. This may be partly owing to the oversimplified empirical relationship between surface albedo and LAI in the Uchijima [45] scheme (Equation (6)), which might have led to an underestimation of surface albedo ($\alpha$) and an overestimation of the net radiation ($R_n$) in late spring and early autumn when the LAI has reached or still remained a high level. In addition, the mismatch between the flux tower footprint (usually hundreds of meters) and the pixel size (5 km) used in this study may be another source of the biases. Also, the uncertainties stemmed from the interpolation of the climatic variables in spatial and temporal aspects should be concerned.

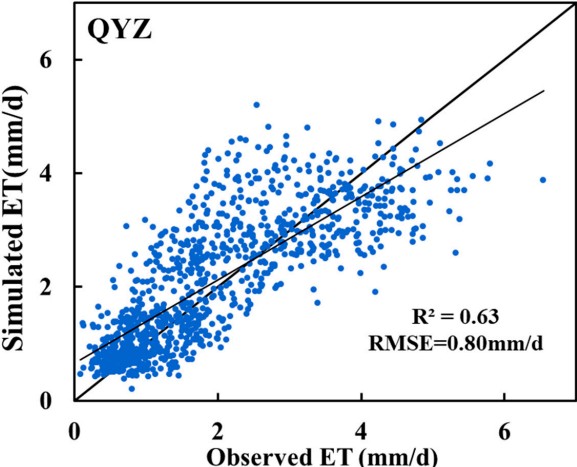

**Figure 6.** Comparisons between the simulated daily evapotranspiration (ET) and measured ET by eddy covariance in Qianyanzhou (QYZ) site from 2003–2005.

The water-balance derived ET at annual and monthly scales were used to verify the simulated basin-scale ET over the GRB, which were calculated as average ET values over the whole basin. As shown in Figure 7, simulated and water-balance derived basin-scale ET were in good agreement, with $R^2$ values being 0.57 and 0.53, and the RMSE being 26.6 mm year$^{-1}$ and 24.8 mm month$^{-1}$ for the annual and monthly series, respectively. Both of the correlations were significant at the 99% confidence level ($p < 0.01$). From 1982 to 2016, the modeled average annual ET over the whole basin was 747.2 mm year$^{-1}$, which was quite close to the water-balance derived average annual ET (744.7 mm year$^{-1}$). However, the water-balance derived annual/monthly ET was generally higher than the simulated annual/monthly ET when the annual ET was above 750 mm year$^{-1}$ or the monthly ET was over 100 mm month$^{-1}$. These biases may be partly attributed to the underestimation of precipitation due to missing of some local storm events in hilly regions of GRB, where the meteorological stations were relatively sparse. In addition, the higher uncertainty of GRACE TWSA in relatively small basins ($\leq$200,000 km$^2$) may be another source of these biases [61,63].

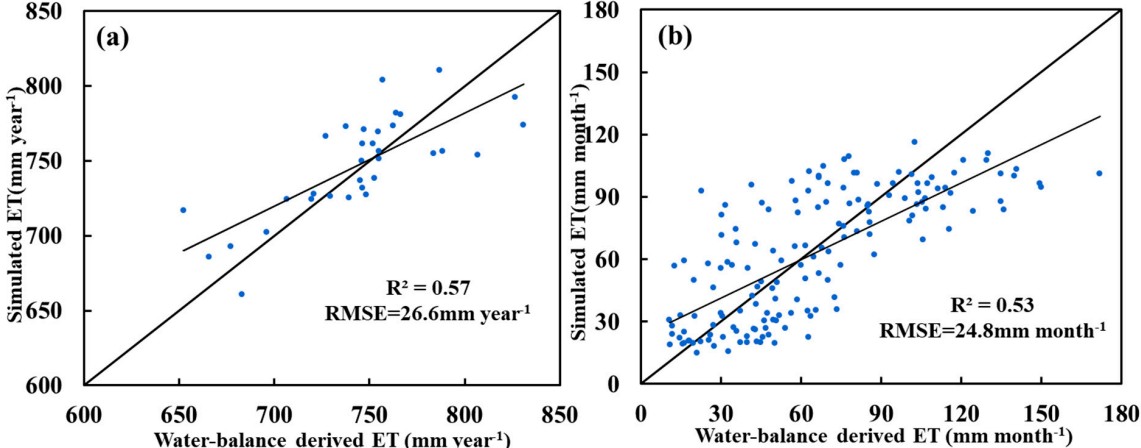

**Figure 7.** Comparisons between the simulated and water-balance-derived ET on (**a**) annual and (**b**) monthly scales over the GRB.

### 3.2. Spatial Patterns of ET, P, and LAI

On a regional scale, the spatial-temporal dynamics of actual ET (ETa) are principally controlled by both precipitation and available energy (denoted by potential ET (ETp) and defined as the FAO reference crop evapotranspiration in this study) and regulated by the vegetation condition (represented by LAI) [71]. Their regulations on the ETa dynamics are associated with the corresponding spatial and temporal scales.

As shown in Figure 8, the average annual ETa, precipitation (P), ETp, and growing season (March to October) LAI (LAIg) illustrated large spatial variabilities over the study basin. The spatial distribution of precipitation showed a remarkable bipolar pattern, with the south-western quadrant being the driest area (<1500 mm year$^{-1}$) and the north-eastern edge being the wettest part (>1700 mm year$^{-1}$). Regulated by patterns of solar radiation and humidity associated with precipitation, ETp exhibited a spatial pattern similar to precipitation but opposite in phase; that is, the highest values were located in the south-western quadrant, and the lowest values were distributed on the north-western edge. It is noted that the distribution of ETp in the north part was also regulated by the terrain; ETp values in western mountainous areas were obviously lower than those in the eastern plain. However, the spatial gradient of ETp is significantly smaller than that of precipitation, being about half of the latter. The spatial pattern of LAIg was clearly associated with topography and land-use types. Higher LAIg values were mainly distributed in the middle and high mountainous areas where the forests are dominant, and lower LAIg values were located in the northern plains and intermountain basins in which the urban areas and cropland are concentrated.

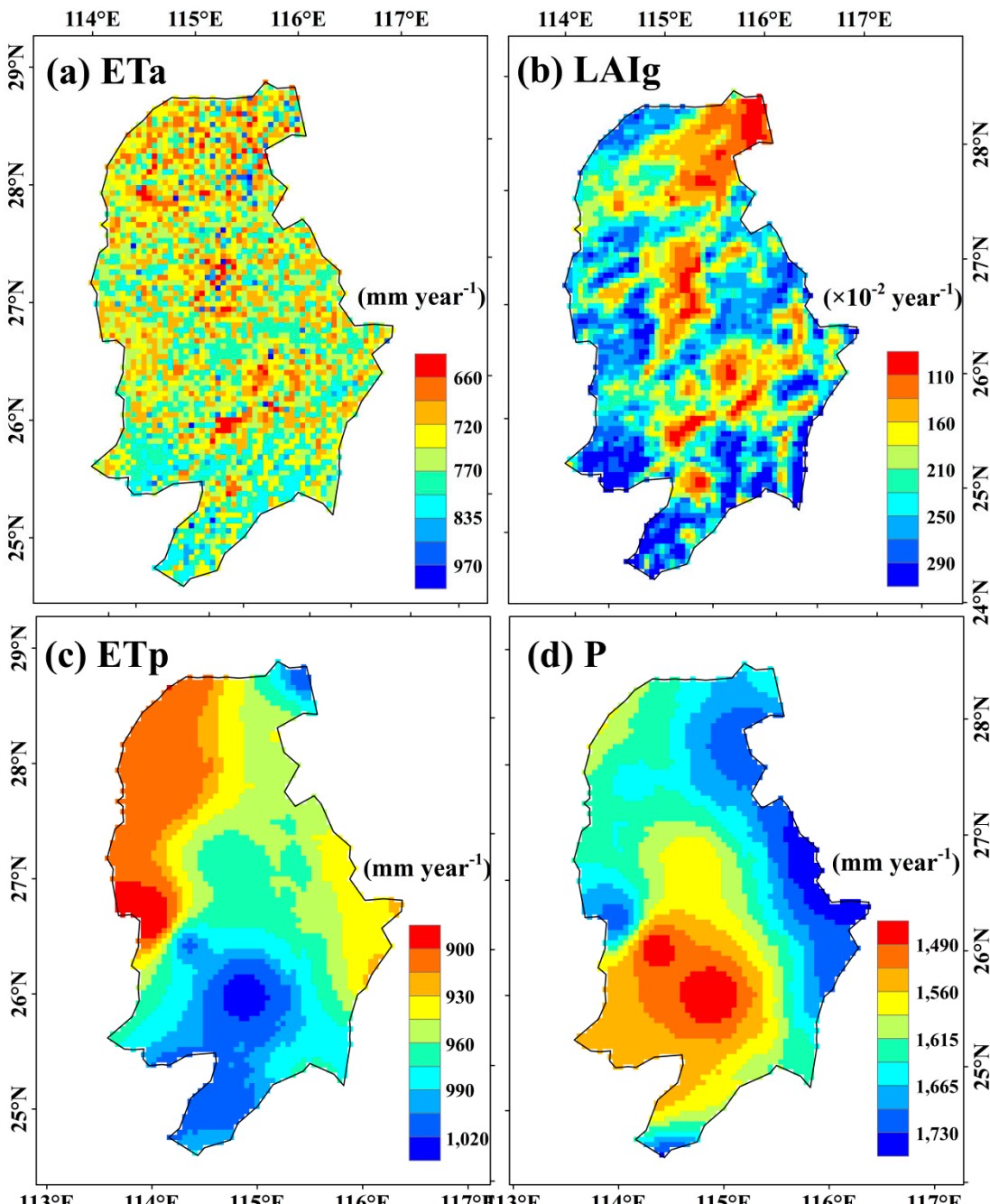

**Figure 8.** Spatial patterns of average annual (**a**) actual ET (ETa), (**b**) growing season LAI (LAIg), (**c**) potential ET (ETp), and (**d**) precipitation (P) over the period 1982–2016 in the GRB.

Due to the comprehensive influence of various factors mentioned above, the spatial pattern of ETa was relatively complicated, varying in all cardinal directions (Figure 8a). However, some identified spatial features still indicate that the spatial variability of ETa mainly followed that of LAIg. Generally, ETa was relatively high in densely vegetated areas in the central and southern regions, ranging from 750 to 900 mm year$^{-1}$. Low ETa values were mainly concentrated in densely populated regions in plains and intermountain basins, being lower than 650 mm year$^{-1}$. In irrigated croplands, however, ETa values were generally higher than those in the surrounding natural vegetation areas, ranging from 650 to 930 mm year$^{-1}$. Over the water bodies, ETa was about 1100 mm year$^{-1}$, roughly equal to the potential evaporation. Averaged over the whole basin, the mean annual ETa was 747.2 mm

year$^{-1}$, accounting for 46% of the corresponding annual precipitation. This means that over a decade, water availability may not be a limiting factor for the atmospheric evaporative demand.

There are noticeable differences in spatially averaged ETa between vegetation types (Figure 9a). It is seen that ETa in cropland is a bit higher than that in the natural vegetation, although the atmospheric water demand (i.e., ETp) is relatively low. This is mainly because irrigation input mitigated the water stress in cropland during dry seasons. Among the natural vegetation types, the evergreen broad-leaved forest (EBF) has the highest ETa rate, which is associated with its relatively large leaf area and low stomatal resistance. The spatial correlation coefficient (*r*) between ETa and LAIg for each vegetation type ranged from 0.43 (for shrub (SH)) to 0.69 (for EBF), being much higher than that for the whole basin (0.24), which does not exclude the physiological differences among the vegetation types. Therefore, it is concluded that the spatial heterogeneity of ETa is principally dominated by the vegetation type and LAI patterns together.

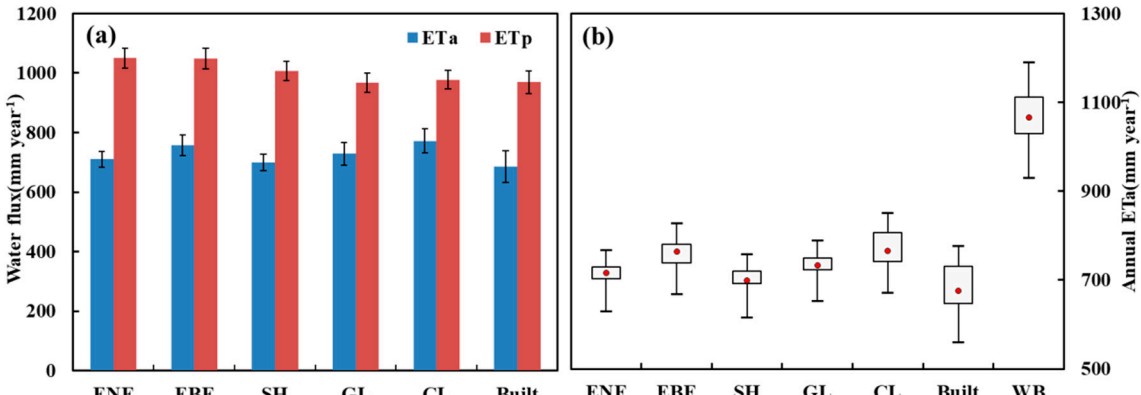

**Figure 9.** Spatially averaged annual ETa and ETp (**a**) and the statistics of ETa temporal variabilities (**b**) over ecosystems within the GRB. Error bars are the spatial standard deviations. Boxplots of each panel illustrate the first and third quantile ranges (box), the median (red circle), and the maximum–minimum range (whiskers) of ETa. (ENF: evergreen needle-leaved forest, EBF: evergreen broad-leaved forest, SH: shrub, GL: grassland, CL: cropland, WB: water body).

### 3.3. Temporal Variations of ET, LAI, and P

#### 3.3.1. Intra-Annual Variations

Averaged spatially over the whole basin, the monthly variations of ET (ETa, ETp, and Ec), LAI, and precipitation all showed strong seasonality (Figure 10a). Except for precipitation, all cases followed a roughly symmetrical single-peak pattern, with the peak occurring in July. This synchronism indicated that the intra-annual variability of water flux was generally dominated by the available energy and canopy leaf area rather than water availability. However, the enlarged deviation between ETa and ETp in July and August implied a limited water supply in the summer months. For the seasonal variations of precipitation, a noticeable asymmetry pattern was observed, with the peak occurring in June and a sharp decline from June to July. Hence, the monthly variations of precipitation in the growing season (March to October) can be divided into two phases, a wet season lasting from March to June and then a dry season lasting from July to October. In the wet season, monthly precipitation is much higher than both ETp and ETa, and only 34.2% of the wet-season precipitation returned to the atmosphere through ETa. In the dry season, however, monthly precipitation quickly decreased to levels comparable to ETp, even lower than the latter in some months, exerting water stress on the atmospheric evaporative demand. During this stage, 76.9% of cumulative precipitation was consumed by ETa, more than twice that during the wet season.

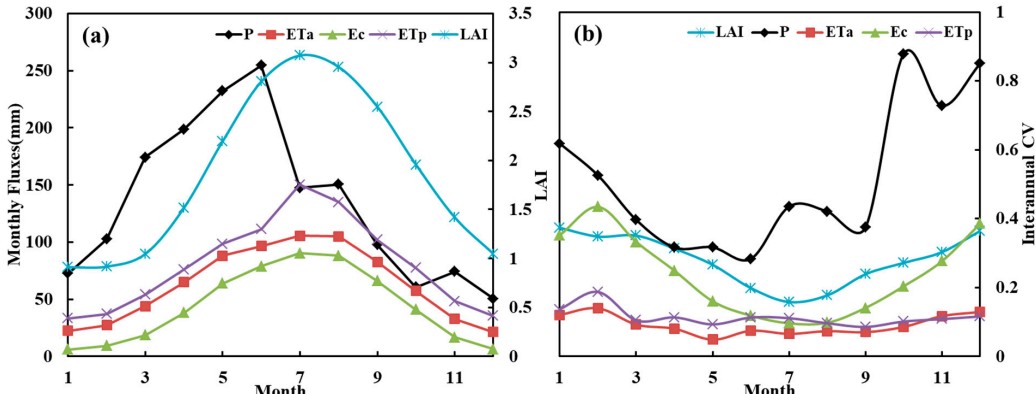

**Figure 10.** Intra-annual variations of (**a**) the quantities and (**b**) the inter-annual coefficients of variation (CV) of ETa, ETp, Ec, P, and LAI.

The inter-annual coefficients of variation (CVs) of monthly ET (ETa, ETp, and Ec), LAI, and precipitation, are presented in Figure 10b. It is shown that the inter-annual CVs of monthly ETa and ETp illustrated similar seasonal variabilities, with a correlation coefficient (*r*) of 0.73 (*p* < 0.01). However, the inter-annual CVs of monthly Ec are more similar to those of monthly LAI, in magnitude and in shape, indicating that the seasonal variability of transpiration is primarily controlled by the vegetation phenology evolution. For each month, the inter-annual CV of monthly precipitation is the highest of CVs of all variables, indicating that the water availability has higher inter-annual oscillation amplitudes than the atmospheric water demand.

### 3.3.2. Inter-Annual Variations

Inter-annual variation of ETa is controlled by climate variability/change and land-use changes. As shown in Figure 11a, ETp exhibited a slightly decreasing trend during the period 1982–2016, which mainly resulted from the offsetting between the negative effects of declining wind speed (−0.1 m decade$^{-1}$, *p* < 0.01) & reduced sunshine duration (−34.8 h decade$^{-1}$, *p* = 0.12) and the positive effect of air warming (0.14 °C decade$^{-1}$, *p* = 0.05) [72]. It is noticed that the ETp decreasing trend tended to level off since 2002, possibly due to the accelerating air warming in the GRB since the 2000s. Simultaneously, ETa showed a slightly increasing trend with lower inter-annual oscillation amplitudes than those for ETp over the study period. The year-to-year changes of ETa and ETp demonstrated a strong correlation (*r* = 0.66, *p* < 0.01), indicating that the inter-annual variations of ETa were principally dominated by available energy. As the main component of ETa, Ec illustrated a remarkably similar inter-annual variation with ETa, but with a much stronger positive tendency. This indicates that the evapotranspiration partitioning have significantly changed due to the rapidly increasing ratio of transpiration to evapotranspiration (Ec/ETa), which is confirmed to be closely correlated with the LAI (*r* = 0.91, *p* < 0.01). Therefore, LAI increase may have exerted a more discernable effect on the long-term process of ETa.

On a monthly scale, ETa mainly increased in the spring and autumn months but decreased in summer months (Figure 11b). The positive tendencies may be attributed to the positive trends of monthly LAI and ETp. However, a decline of ETp closely linked to the negative trend of sunshine duration may lead to the downward tendency of ETa in June and August. Owing to the positive effects of increased LAI and precipitation on transpiration, Eta, and ETp in summer months did not change proportionally and even show the opposite trends. The inter-annual variabilities of ETa over ecosystems were displayed by boxplots shown in Figure 9b. It is illustrated that the temporal variability of ETa (i.e., the ETa range between quartile lines) in cropland was much higher than those in natural vegetation communities, close to that in the water body, possibly owing to the relatively low stomatal resistance and sufficient water supply. Among natural vegetation communities, EBF has the highest ETa inter-annual range, exhibiting a more sensitive eco-hydrological response to climate variations.

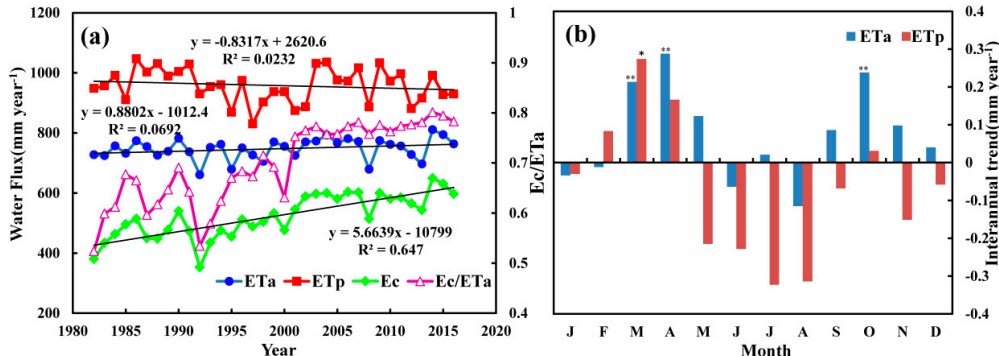

**Figure 11.** Inter-annual variations of annual ETa, ETp, Ec, and Ec/ETa (**a**) and inter-annual trends of monthly ETa and ETp (**b**) from 1982 to 2016 (significance levels: * is $p < 0.05$; ** is $p < 0.01$).

Based on the Budyko framework, the dryness index (ETp/P) and evaporative index (ETa/P) of each year for every vegetation type were calculated and plotted against the Budyko curve [73] to investigate the long-term hydrological responses of different ecosystems to the climatic variability [74]. The location of a particular ecosystem in Budyko's space illustrates a reference condition of the annual water balance as a function of the climatic conditions during any particular year [75].

As shown in Figure 12a, Budyko points of each ecosystem are all below the energy limit line and concentrated around the Budyko curve. This indicated that the long-term evolution of hydrological condition was mainly controlled by energy availability and that the hydrological response of each ecosystem to climate variability has been basically stable so far. It is noticed that the evaporative index of cropland was generally higher than that of other ecosystems, owing to the irrigation input and agronomic management. An interesting finding was that the evaporative index over cropland presented a weakly positive trend (0.007 decade$^{-1}$, $r = 0.08$) during the study period, whereas the evaporative index over the rest of the ecosystems, as well as over the GRB, all demonstrated a slightly negative trend ($-0.015$–$-0.003$ decade$^{-1}$, $r = -0.21$–$-0.04$), in accordance with the tendency of dryness index ($-0.019$–$-0.012$ decade$^{-1}$, $r = -0.16$–$-0.10$). This phenomenon may be owing to the sufficient water supply and increasing LAI in the cropland and confirms the prominent role of agricultural practices (such as large-scale irrigation) in the long-term evolution of the water cycle [76].

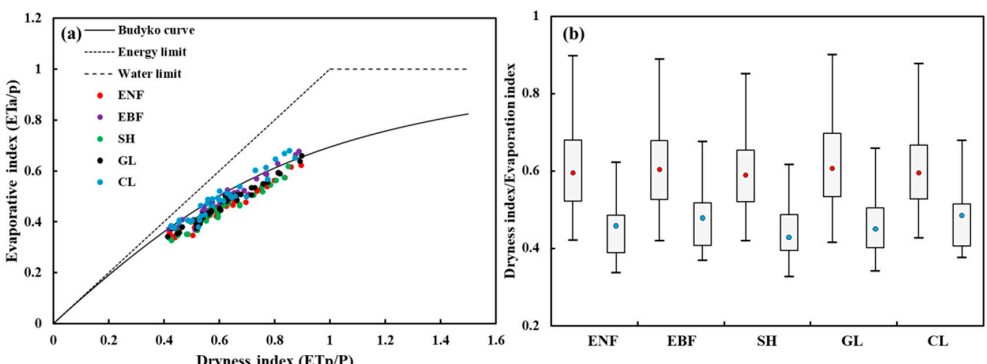

**Figure 12.** Reference hydroclimatic conditions of five ecosystems in Budyko's space (**a**) and the statistics of the temporal variabilities of the dryness index (ETp/P) and evaporative index (ETa/P) for each ecosystem (**b**) during the period 1982–2016. In Figure 12b, the boxplots with red circles show statistics of the dryness index, and those with blue circles show statistics of the evaporative index.

Figure 12b shows the ranges of the dryness index and evaporative index during the study period for each ecosystem. It is found that the evaporative index for evergreen needle-leaved forest (ENF) and shrub (SH) was slightly lower than that for the rest of the ecosystems, whereas there were no distinct differences in climate conditions (dryness index) between ecosystems. This is closely related to their physiological characteristics and the higher stomatal conductance.

### 3.3.3. Temporal Trend of ETa in Relation to P, ETp, and LAI

The inter-annual trends of ET, precipitation, and LAIg showed quite different spatial patterns (Figure 13). Precipitation exhibited an upward trend almost throughout the basin, although significant trends were identified in only 5% of the whole basin ($p < 0.05$). A noticeable wetting trend (>15 mm year$^{-2}$ for precipitation) was identified in the western mountainous areas where a large amplitude of dry-wet transition from the 1980s to 2000s was observed. Similar to the patterns of average annual ETp and precipitation, ETp generally demonstrated a temporal tendency that is opposite to precipitation trend. ETp was significantly decreasing mainly in the western part (19.6% of the whole basin, $p < 0.05$), corresponding to the noticeable wetting trend in this area. LAIg was significantly increasing in almost the whole basin (97.8% of the GRB, $p < 0.05$), revealing remarkable vegetation greening trend throughout the GRB. The spatial pattern of the LAIg trend is clearly related to the topography and land-use types. The LAIg trend in the mountainous areas is evidently higher than that in plain areas, agreeing well with the pattern of MRL and GFG projects, in which the farmland in hilly or mountainous areas was designed to be converted to forestland, and the existing forest land was improved and conserved.

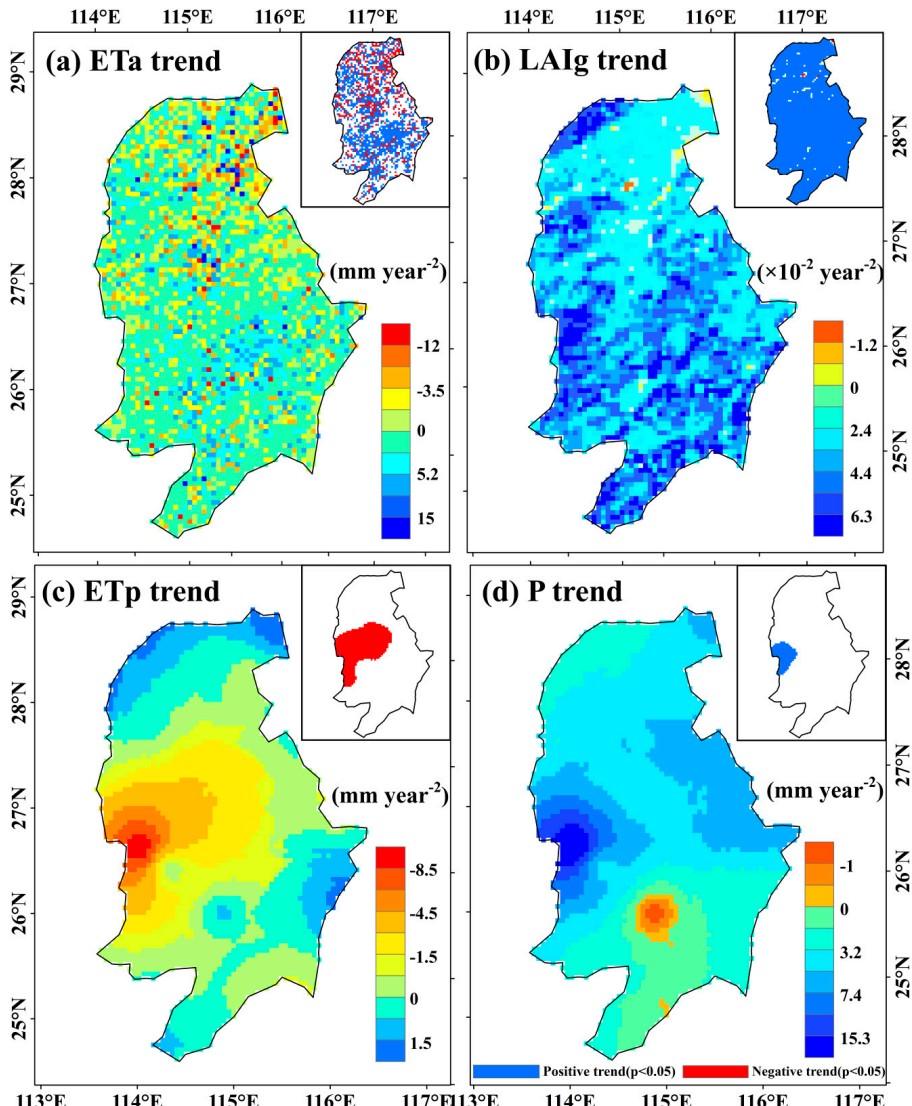

**Figure 13.** Spatial patterns of inter-annual trends of ETa (**a**), LAIg (**b**), ETp (**c**), and P (**d**) over the GRB during the period 1982–2016. Locations with statistically significant trends ($p < 0.05$) are shown in the sub-panels, where positive trends are marked in blue and negative trends in red.

Compared with trends of other variables, the trend of ETa showed more significant spatial variability over the basin (Figure 13a), displaying a comprehensive response of water vapor flux to both climate variabilities and human activities. ETa significantly increased and decreased in 37.2% and 16.8% of the whole basin, respectively ($p < 0.05$); the former was mainly concentrated in and around the mountain basins in the upper GRB, and the latter was more scattered in the lower plains of the GRB. Averaged over the whole basin, the inter-annual trend of ETa was 0.88 mm year$^{-2}$ from 1982 to 2016. However, averaged spatially over ecosystems, ETa trends were remarkably different, ranging from −4.35 to 2.29 mm year$^{-2}$ ($r = -0.79$–0.53 Figure 14a). ETa in the evergreen broad-leaf forest and cropland was increasing while the trend in other ecosystems was negative. These inconsistencies may be owing to differences in LAI trends and physiological characteristics (such as canopy stomatal conductance) among vegetation types.

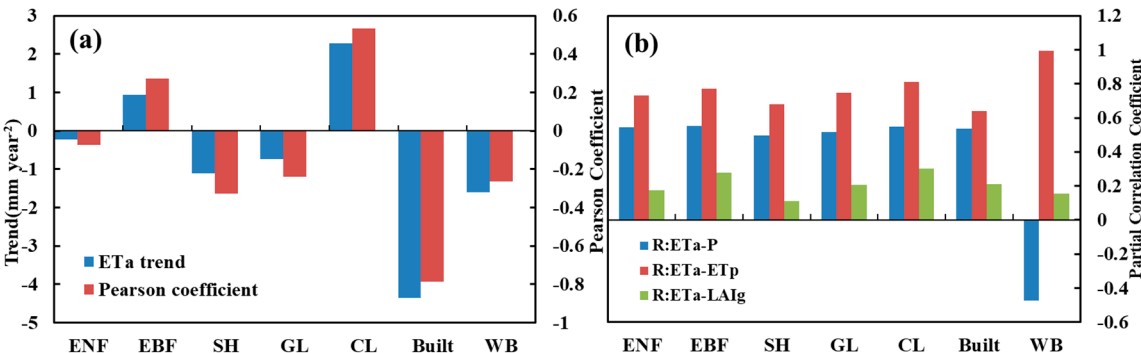

**Figure 14.** Spatially averaged annual ETa trends and their Pearson coefficients with time for the ecosystems (**a**); partial correlation coefficients between ETa and its driving forces for each ecosystem (**b**).

The partial correlation coefficients (R) were calculated to investigate the driving effects of precipitation, ETp, and LAI on the ETa trend over different ecosystems (Figure 14b). It is found that the R-value between ETa and ETp ($R_{ETa-ETp}$) was the highest for all ecosystems, confirming the dominant effects of available energy on the year-to-year changes of ETa. Except for the water body, $R_{ETa-P}$ was positive for all the ecosystems, with little differences among vegetation types. Compared with $R_{ETa-ETp}$ and $R_{ETa-P}$, $R_{ETa-LAIg}$ values were much lower for all the ecosystems, mainly because the trend components were removed from all the time series. However, it can be inferred that LAIg has had a large impact on the ETa trend since the trend of LAIg was much more significant than those of the ETp and precipitation. The opposite ETa trends to those for other ecosystems may be partly attributed to the higher $R_{ETa-LAIg}$ values of EBF and CL than those of others.

*3.4. Sensitivity to Inter-Annual Variability in Climate and LAI*

Sensitivity analysis was performed on a grid basis to further determine the quantitative relationship between ETa and its driving factors. To avoid collinearity between climatic variables, only three independent variables (precipitation, air temperature, and sunshine duration) were used to represent water and energy availabilities. LAI was used to characterize the vegetation change. Partial correlation analysis indicated that there were only weak correlations among the selected variables ($r = -0.33$–0.34, $p = 0.054$–0.687), and the collinearity is unlikely to have a major impact on the sensitivity analysis.

Figure 15 shows the inter-annual sensitivities of annual ETa to climate and LAI variabilities. It is found that there were both spatial and inter-ecosystem differences in sensitivities. As shown in Figure 15a, sensitivities of annual ETa to precipitation (S (ETa, P)) were generally positive, but the magnitude was relatively low (most values ranged from −0.05 to 0.16). High S (ETa, P) values were mainly concentrated in the upstream mountainous areas where the precipitation was relatively low. Also, S (ETa, P) was relatively high in plain areas in the middle and lower reaches, indicating that there was a certain degree of water stress in plain ecosystems during the dry season.

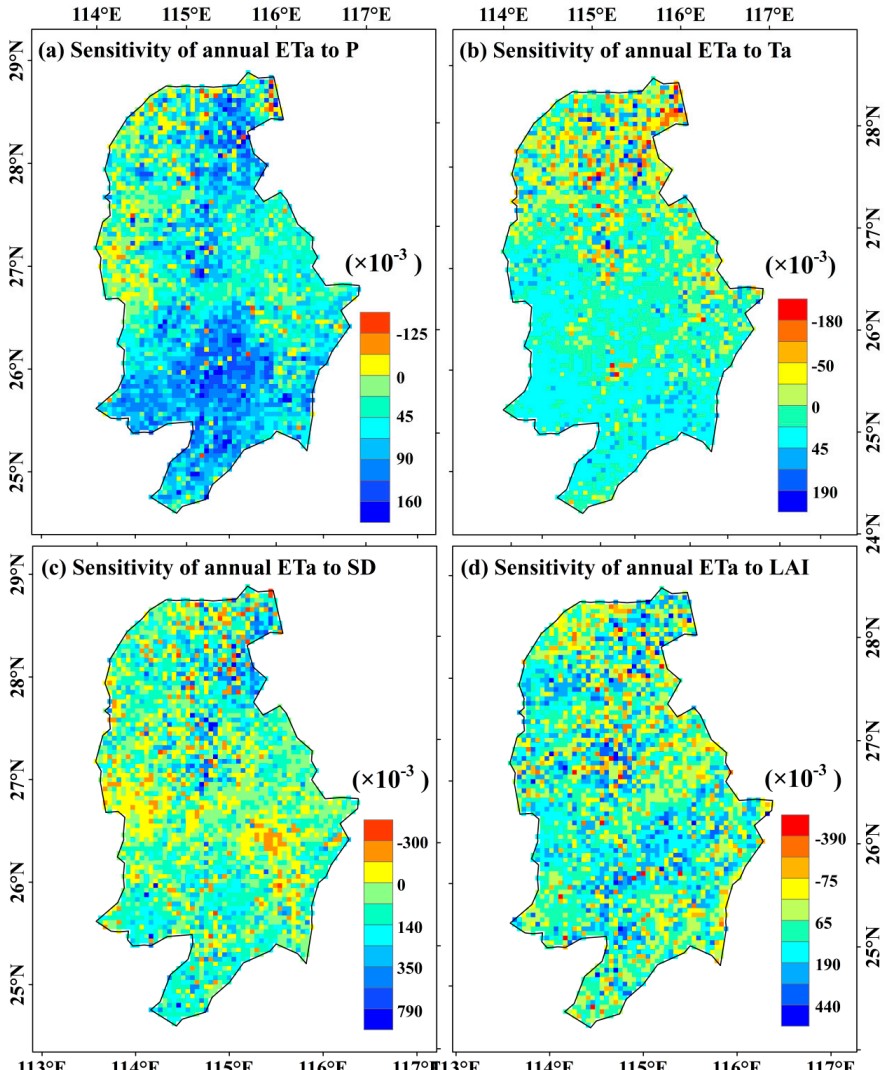

**Figure 15.** Sensitivities of annual ETa to the climatic variables and leaf area index. (**a**) Sensitivity of annual ETa to P; (**b**) Sensitivity of annual ETa to Ta; (**c**) Sensitivity of annual ETa to SD; (**d**) Sensitivity of annual ETa to LAI.

Sensitivities of annual ETa to air temperature (S (ETa, Ta)) illustrated large spatial variability (Figure 15b). The positive values were dominant in upstream mountainous areas, whereas both the positive and negative values were distinctive in the downstream plain. Air warming will enhance the atmospheric water vapor deficit. However, a higher temperature during the dry season may result in the closure of leaf stoma, especially in the plain area.

Sensitivities of annual ETa to sunshine duration (S (ETa, SD)) were generally positive with relatively high magnitude. High S (ETa, SD) values were concentrated in the cropland in middle and lower reaches where water supply was sufficient. The average S (ETa, SD) over the whole basin was 0.083, much higher than that for precipitation (0.057) and for air temperature (0.011), confirming the dominant role of energy availability in driving water vapor flux in this humid basin.

Sensitivities of annual ETa to LAI (S (ETa, LAI)) were generally positive and showed a spatial pattern closely linked to the vegetation coverage and ecosystem types. S (ETa, LAI) is positively related to the canopy leaf area and is regulated by water availability. Namely, areas with high values correspond to low vegetation coverage regions or cropland with irrigation input. Therefore, it can be inferred that the eco-hydrological response of the farmland ecosystem to vegetation greening may be more remarkable than that of the natural ecosystem.

Based on the sensitivities of the annual ETa to climate and LAI, contributions of climate change and vegetation greening to ETa trend were estimated over different ecosystems and over the GRB (Figure 16). It is found that climate change contributed negatively to ETa trends in all the ecosystems, whereas the contributions of vegetation greening were positive for all of the vegetation ecosystems. The negative effects of LAI in Built-up land and water bodies are owing to the expansion of urbanization, which resulted in deforestation and devegetation in urban areas and reduced water bodies around cities. The contribution of vegetation greening in EBF was much higher than that in other natural vegetation ecosystems. This may be related to the more distinct leaf area growth and lower stomatal resistance in EBF. As a result, the negative effect of climate change on ETa has been eventually offset by the positive effect of the increasing leaf area. It is noticed that the effect of climate change on ETa in cropland was slight and neglectable. This may be closely linked to agronomic management (irrigation, fertilization, and new cultivars adoption), which increased crop production and water consumption. Averaged over the whole basin, contributions of the climate change and vegetation greening on the ETa trend were $-0.48$ mm year$^{-2}$ ($-54.8\%$) and $1.36$ mm year$^{-2}$ ($154.8\%$), respectively. It is clear that vegetation greening was the dominant driver for the long-term trend of ETa in the GRB over the past several decades.

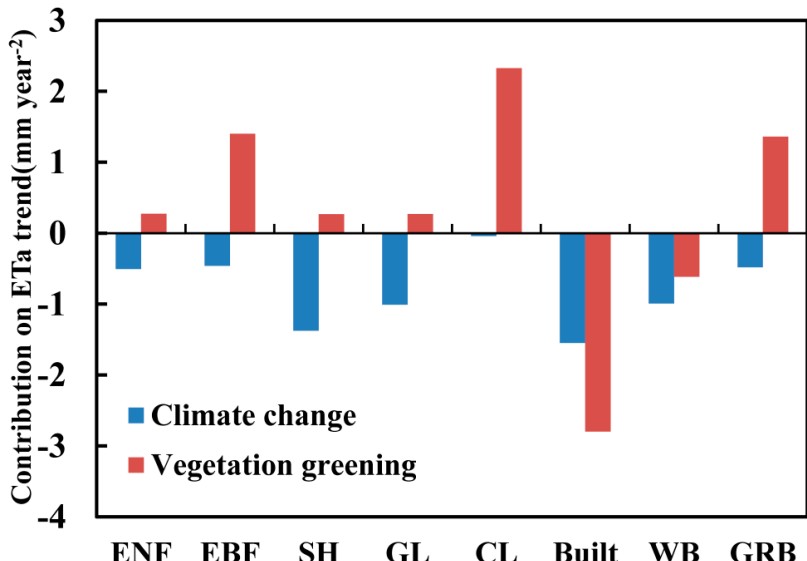

**Figure 16.** Contributions of climate change and vegetation greening to ETa trends over ecosystems and over the GRB.

## 4. Conclusions and Discussion

Over the past several decades, climate change and human activities have largely altered the hydrological regimes in the Basin of Poyang Lake (PLB), the largest freshwater lake of China, posing a potential threat to water resources sustainability and ecosystem security. In this study, a process-based ET model in conjunction with the GLASS LAI dataset was used to characterize the spatial-temporal pattern of evapotranspiration from 1982 to 2016 over the largest sub-basin of PLB-Gan River Basin (GRB). Validated with eddy covariance flux and water-balance derived ET, the model is proved to be reliable. It should be pointed out that the daily-basis ET predictions were validated with only one flux tower, and thus predictions in only one ecosystem were verified, leaving uncertainties in results based on the ecosystem type to some extent. However, we expected that these limitations could not have significantly influenced the final results since most parameters (including those related to vegetation type) in the model are more physical rather than empirical.

Simulation results showed that the actual annual ET (ETa) weakly increased with an annual trend of 0.88 mm year$^{-2}$ from 1982 to 2016 over the GRB, along with a slight decline in annual potential ET (ETp). The increases in ETa mainly occurred in spring and autumn, while ETa decreased in

the summer, owing to the decline in sunshine duration. On an ecosystem basis, however, only the evergreen broad-leaf forest and cropland presented a positive ETa trend, while the rest of the ecosystems demonstrated negative trends of the ETa, being consistent with that of ETp. Both correlation analysis and sensitivity analysis revealed a close relationship between ETa inter-annual variability and the availability (represented by ETp). Correlation analysis illustrated that contributions of climate change and vegetation greening on the ETa trend were $-0.48$ mm year$^{-2}$ ($-54.8$%) and $1.36$ mm year$^{-2}$ ($154.8$%), respectively. Climate change had a negative impact on the ETa trend over the GRB. However, the negative effects have been offset by the positive effects of vegetation greening, which mainly resulted from the large-scale revegetation in forestland and agricultural practices in cropland. It is concluded that human activities were the main drivers of the long-term evolution of ETa over the GRB. Our results agree well with the results of Liu et al. [77], who studied the long-term evolution of water cycle over eastern China during the past decades and concluded that climate change dominated the inter-annual variability of ET, while land-use change exerted more discernable effects on the hydrological process in the long run.

Long-term variations of ETa inevitably have an impact on the regional water and energy balance, thus affecting other aspects of the hydrological cycle, including streamflow. On an annual scale, our results showed a long-term positive trend in ETa but a weakly negative trend in the evaporative index due to the more discernable increase in the precipitation. This negative trend in the evaporative index was verified by the observed increase in the runoff coefficient over the last several decades [78]. This confirmed the dominant role of precipitation in the long-term variations of annual streamflow in the study area [8,11]. On the seasonal and monthly scales, changes in ETa may have shown more distinct effects on the streamflow. For example, an increase of ETa in spring may have decreased the streamflow since the precipitation did not change significantly, thus reducing the flood potential in this season [7]. However, a decrease of ETa along with an increase of precipitation in August have resulted in a significant increase in the streamflow [11], thus alleviating the water stress [7], while a significant increase of ETa together with a decrease of precipitation in October may have decreased the streamflow and further enhanced the drought impacts.

Our study also revealed a significant change in the evapotranspiration partitioning due to the rapidly increasing ratio of transpiration to evapotranspiration (Ec/ETa). Over the study period, Ec/ETa in the GRB displayed a remarkable increase from 0.52 in 1982 to 0.78 in 2016, which is closely related to the LAI increase. The multi-year average Ec/ETa over the GRB was 0.70, which agreed well with the measured Ec/ETa of 0.72–0.77 in the subtropical forest sites of eastern China [79]. A higher Ec/ETa level suggests a higher proportion of biological flux and a lower proportion of physical flux between terrestrial ecosystems and the atmosphere [80]. This means that the GRB may have had a higher hydrological resilience in response to the projected increase in climatic extremes, especially the drought since vegetation can respond to the expected climatic and hydrological anomalies through adjusting the leaf stomatal conductance [81,82].

Our study also indicates that an ecosystem-basis analysis is necessary to quantify the long-term evolution of the hydrological cycle in response to environment changes. This study can improve our understanding of the interactive effects of climate change and human activities on the long-term evolution of water cycles.

**Author Contributions:** Conceptualization, M.B. and B.S.; methodology, M.B., X.S. and S.M.; software, M.B.; validation, S.M., L.H. and Q.Q.; formal analysis, M.B.; writing—original draft preparation, M.B.; writing—review and editing, M.B., B.S. and X.S.; supervision, B.S.; funding acquisition, S.M.

**Funding:** This research was funded by the National Natural Science Foundation of China (No. 51679185).

**Acknowledgments:** We thank to all the data providers. We also appreciate the editors and anonymous reviewers for their constructive comments and suggestions, which led to significant improvements in the manuscript.

**Conflicts of Interest:** The authors declare no conflict of interest.

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
