# Peer review of "Multi-Temporal Variabilities of Evapotranspiration Rates and Their Associations with Climate Change and Vegetation Greening in the Gan River Basin, China"

_water, doi:10.3390/w11122568_

Round 1

Reviewer 1 Report

This is my third review of this manuscript. After having read through this version of the manuscript thoroughly, I am happy to report that my extensive comments from the first two submissions have been addressed. The study appears to have been carefully done and has been substantially improved in revision. In my opinion, this manuscript is ready for publication. Congratulations on an interesting and informative piece of work.

Author Response

Dear reviewer,

We sincerely appreciate your positive comments of our manuscript and suggestion about the paper publication. Thanks very much for your great efforts to review the manuscript, which has certainly benefited from your insightful revision suggestions.

Sincerely,

Meng Bai (First author of Water-639010)

Reviewer 2 Report

This is a paper of general good quality. It addresses a topic of scientific and practical interest: the spatial and temporal dynamics of evapotranspiration in relation to climate change and human activities in

 the largest sub-basin of Poyang Lake catchment (in China). The paper is based on appropriate and modern methods, clearly described. The results indicates that an ecosystem-basis analysis is necessary to quantify the long-term evolution of hydrological cycle in response to environment changes. They are originals and interesting especially at the national and regional level.

Several minor suggestions are made below.

In introduction I suggest to highlight the novelty/originality of the paper compared to previous studies/researches and the interest.

In Abstract, please replace the semicolon (;) with comma (,), because the use of the semicolon is not justified.

Please  check and correct several bibliographic citations in the text, because there is not concordance between the quotation in text and the References, as follows:

Row 52: In the reference list is Sun et al, not Son, Chen (reference 12).

Row 55: In the reference list is Ye et al.,  not Ye, Zhang (reference 13).

Row 58: In the reference list is Zhang et al , not Zhang, Liu (reference 14).

Row 66: In the reference list is Huang et al., not Huang, Shau (reference 15).

Row 71: In the reference list is Guo, H., Q. Hu, and T. Jiang,  not Guo, Hu (reference 6).

Row 71: In the reference list is Tang et al., not Tang, Cai (reference 9).

Row 169: In the reference list is Liu et al., not Liu, Shao (reference 39)

Row 257: In the reference list is Wan et al., not Wan, Zhang (reference 59)

Row 585: In the reference list is Liu et al., not Liu, Tian (reference 70)

Other small corrections:

Row 119: It is recommended to use/mention the multiannual average discharge, not only the volume.

Row 176: to use point instead of ; after environmental conditions

Row 177: to use point instead of ; after surface layer

Row 177: to use point instead of ; after canopy height

Row 185: to use comma instead of ; after Hornberger [44]

Row 247: to delete one point after spatial domain

Author Response

Dear reviewer,

We appreciate the constructive comments and suggestions you provided. They are really helpful for improving our manuscript. We have carefully taken into account these comments/suggestions and tried our best to incorporate the suggested changes into the manuscript. The following are your comments/suggestions and our responses.

Comment 1: In introduction I suggest to highlight the novelty/originality of the paper compared to previous studies/researches and the interest.

Response 1: In introduction, we have added more references and citations which mainly refer to the eco-hydrological responses of ecosystems to climate variabilities and vegetation dynamics. We have put more emphasis on the diversity responses of different ecosystems to the environmental changes due to differences in physiological structures among vegetation types. Also, the use of GRACE data in the derivation of the water-balance ETa has been emphasized.

Comment 2: In Abstract, please replace the semicolon (;) with comma (,), because the use of the semicolon is not justified.

Response 2: Done.

Comment 3: Please check and correct several bibliographic citations in the text, because there is not concordance between the quotation in text and the References, as follows:

Row 52: In the reference list is Sun et al, not Son, Chen (reference 12).

Row 55: In the reference list is Ye et al., not Ye, Zhang (reference 13).

Row 58: In the reference list is Zhang et al , not Zhang, Liu (reference 14).

Row 66: In the reference list is Huang et al., not Huang, Shau (reference 15).

Row 71: In the reference list is Guo, H., Q. Hu, and T. Jiang,  not Guo, Hu (reference 6).

Row 71: In the reference list is Tang et al., not Tang, Cai (reference 9).

Row 169: In the reference list is Liu et al., not Liu, Shao (reference 39)

Row 257: In the reference list is Wan et al., not Wan, Zhang (reference 59)

Row 585: In the reference list is Liu et al., not Liu, Tian (reference 70)

Response 3: We have modified the format of the citations and reference list according to the guideline of the journal.

Comment 4: Other small corrections:

Row 119: It is recommended to use/mention the multiannual average discharge, not only the volume.

Row 176: to use point instead of ; after environmental conditions

Row 177: to use point instead of ; after surface layer

Row 177: to use point instead of ; after canopy height

Row 185: to use comma instead of ; after Hornberger [44]

Row 247: to delete one point after spatial domain

Response 4: We have carefully checked the whole text and corrected some grammatical and punctuation errors.

Sincerely,

Meng Bai (First author of Water-639010)